

# Sea ice led to poleward-shifted winds at the Last Glacial Maximum: the influence of state dependency on CMIP5 and PMIP3 models

Louise C. Sime[1], Dominic Hodgson[1], Thomas J. Bracegirdle[1], Claire Allen[1], Bianca Perren[1], Stephen Roberts[1], and Agatha M. de Boer[2]

[1]British Antarctic Survey, Cambridge, U.K.
[2]Bert Bolin Centre for Climate Research, Department of Geological Sciences, Stockholm University, Stockhom, Sweden

*Correspondence to:* Louise C. Sime (lsim@bas.ac.uk)

**Abstract.** Latitudinal shifts in the Southern Ocean westerly wind jet could drive changes in the glacial to interglacial ocean $CO_2$ inventory. However, whilst CMIP5 model results feature consistent future-warming jet shifts, there is considerable disagreement in deglacial-warming jet shifts. We find here that the dependence of pre-industrial (PI) to Last Glacial Maximum (LGM) jet shifts on

PI jet position, or state dependency, explains less of the shifts in jet simulated by the models for the LGM compared with future-warming scenarios. State dependence is also weaker for intensity changes, compared to latitudinal shifts in the jet. Winter sea ice was considerably more extensive during the LGM, along with state dependence, changes in surface heat fluxes due this sea ice change probably had a large impact on the jet. Models which both simulate realistically large expansions in

sea ice and feature PI jets which are south of 50°S, show an increase in wind speed around 55°S, and can show a poleward shift in the jet between the PI and the LGM. However models with the PI jet positioned equatorwards of around 47°S do not show this response: the sea ice edge is too far from the jet for it to respond. In models with accurately positioned PI jets; a $+1°$ difference in the latitude of the sea ice edge tends to be associated with a $-0.85°$ shift in the 850 hPa jet. However,

it seems that around 5° of expansion of LGM sea ice is necessary to hold the jet in its PI position. Since the Gersonde et al. (2005) data supports an expansion of more than 5°, this result suggests that a slight poleward shift and intensification, was the most likely jet change between the PI and the LGM. Without the effect of sea ice, models simulated polewards shifted westerlies in warming climates and equatorward shifted westerlies in colder climates. However, the feedback of sea ice

counters and reverses the equatorward trend in cooler climates so that the LGM winds were more likely to have also been shifted slightly poleward.





## 1 Introduction

The concentration of $CO_2$ in the atmosphere decreases by ~90 parts per million between warm interglacial and cold glacial climate states, due to oceanic storage of the excess carbon. Mechanisms behind this enhanced ocean storage are still unresolved. One hypothesis invokes latitudinal shifts in the Southern Ocean westerly wind belt. An equatorward, or weaker, westerly wind jet could suppress deep water ventilation, leading to carbon becoming trapped in cold dense waters (Toggweiler et al., 2006; Sigman et al., 2010; Denton et al., 2010).

The evidence in favour of jet shifts driving increased glacial oceanic carbon storage though direct physical and biological carbon pumps is weak. Authors including Menviel et al. (2008); Tschumi et al. (2008); d'Orgeville et al. (2010); Lee et al. (2011) have investigated the effect that wind jet shifts have on ocean circulation during the Last Glacial Maximum (LGM) using numerical models. In one of the most complete recent studies, Völker and Köhler (2013) simulated the impact of jet shifts with a full ocean general circulation model (MITgcm), spun up to simulate LGM conditions, and used a dissolved inorganic carbon package (MITgcm Group, 2013) to simulate carbon changes. They found small net effects on atmospheric carbon, with a rise of only 3 to 9 ppm $CO_2$ under both a north and a southward $10°$ shift of the surface jet. These results are similar to those obtained using some simpler ocean models (Menviel et al., 2008; Tschumi et al., 2008; d'Orgeville et al., 2010). However, the effects on ocean circulation and biology are complex and non-linear, with competing effects from physical and biological carbon pumps. Thus it is difficult to know if these model-based studies are sufficiently accurate to constrain the $CO_2$ impact of a specified wind shift. Thus whilst most, though not all (*e.g.* Lee et al., 2011), ocean and carbon modelling results do not support the idea that shifts in the westerly wind belt played a dominant role in coupling atmospheric $CO_2$ rise and global temperature there is, as yet, no definitive answer to this question.

Jet shifts have been proposed to modify other aspects of the climate-$CO_2$ system. Iron rich dust borne by Southern Hemisphere winds is thought to increase Southern Ocean productivity (Kohfeld et al., 2005). Lamy et al. (2014) show that large-scale Southern Hemisphere climate forcings, likely wind related, enhanced cold glacial period dust mobilization in Australia, New Zealand, and Patagonia. Ferrari et al. (2014) hypothesise a Southern Ocean dividing latitude between negative and positive buoyancy forcing at the edge of the summer sea ice edge, with knock-on impacts for ocean dynamics. Völker and Köhler (2013) results suggest that if the atmospheric jet shifts poleward, summer sea ice extends, likely due to enhanced heat loss to the atmosphere. Thus both dust and buoyancy forcing may provide an additional means for jet changes to influence glacial to interglacial climate shifts.

A wide range of paleodata has been interpreted as evidence for glacial to interglacial jet shifts. This data includes proxies, or direct measurements of, terrestrial moisture, dust deposition, sea surface temperatures, and ocean productivity. Kohfeld et al. (2013) find that purely based on these paleodata, one can hypothesise a variety of wind change scenarios including: no change, a southward shift, and a



northward shift. It remains an extraordinarily difficult task to constrain glacial to interglacial jet shifts

and intensifications based on data alone (Hodgson and Sime, 2010). Whilst Sime et al. (2013) find that the moisture change paleodata can be accurately modeled under a no jet shift scenario, Kohfeld et al. (2013) suggest that an equatorward jet shift, or intensification, could also be consistent with the majority of the paleodata. Efforts to help solve this jet change problem using GCMs have benefitted from the fifth Coupled Models Intercomparison Project (CMIP5), specifically the third Paleoclimate

Models Intercomparison Project (PMIP3), and its predecessor PMIP2 (Braconnot et al., 2007, 2012; Taylor et al., 2011). PMIP2 and PMIP3 have provided ensembles of LGM and PI climate simulations, where each model is run under the same boundary conditions, permitting intermodel comparisons and insight into crucial wind change mechanisims (*e.g.* Rojas, 2013; Chavaillaz et al., 2013; Sime et al., 2013).

Existing analyses of PMIP2 and PMIP3 LGM simulation ensembles show considerable inter-model disagreement in PI to LGM Southern Hemisphere jet changes (Rojas, 2013; Chavaillaz et al., 2013). This is despite the fact that nearly all CMIP5 models exhibit a poleward shift, and all models a strengthening, of the surface jet from 1900 to 2100 (Bracegirdle et al., 2013). Indeed Chavaillaz et al. (2013) find that future-warming scenario RCP4.5 (Representative Concentration Pathway 4.5)

shifts in the 850 hPa jet can be largely explained by tropospheric temperature differences between the southern high latitudes and the tropics. Bracegirdle et al. (2013) and Kidston and Gerber (2010) examine another aspect: state dependency for Southern Ocean jet shifts and intensity changes, where state dependency is defined as the dependence of jet shifts on the start jet position. Bracegirdle et al. (2013) find that for some oceanic sectors, particularly the Pacific, the starting position of the jet (state

dependence) explains more than 85% of the jet shift variance found between the different CMIP5 future-warming scenario simulations. This implies that the start latitude of the jet is potentially a strong contender as an explanation for CMIP5-PMIP3 inter-model jet shift differences. Additionally, whilst tropical temperature changes dominate the future-warming wind changes, high latitude temperature changes are as significant to the winds during the deglacial-warming (Chavaillaz et al.,

2013). Sea ice, and potentially also Antarctic ice sheet changes, are thus highlighted also as being particularly significant for the accurate jet simulations (Chavaillaz et al., 2013; Sime et al., 2013). Here we investigate past-cooling LGM state dependency, sea ice, and changes in the Southern Ocean westerly wind jet using CMIP5-PMIP3 output.

## 2 Data: CMIP5-PMIP3 simulations

CMIP5-PMIP3 pre-industrial (PI) and Last Glacial Maximum (LGM) simulations are run with full dynamic ocean and sea ice models. The LGM simulations all follow the PMIP3 protocol ($https : //wiki.lsce.ipsl.fr/pmip3/doku.php/pmip3 : design : 21k : final$):orbital parameters are set to their 21 000 yr ago values; concentration of atmospheric greenhouse gases are set to 185 ppm for





$CO_2$; 350 ppb for $CH_4$; and 200 ppb for $N_2O$. All models use the PMIP3 LGM ice sheet or ICE5.2G
ice sheet configurations (Chavaillaz et al., 2013). Simulations are run for long enough to allow the
atmosphere and ocean to reach quasi-equilibrium (Braconnot et al., 2012; Rojas, 2013).

Models and simulations are shown in Table 1. For some models more than one realization is
available (*i.e.* the same model is run more than once with the same forcing). Where more than one
realization is available (indicated in Table 1), the mean of those realizations is used. Additionally,
some models (*e.g.* GISS-E2-R-p150 and GISS-E2-R-p151) differ only slightly in their physics. In
this case, as above, we use a mean of these model simulations. This yields a total of nine independent
PMIP3 simulations.

### 2.1 Southern Ocean wind jet diagnostics

The choice of Southern Ocean jet diagnostic can influence apparent glacial to interglacial wind
change results. Previous authors have used: surface winds (Kim et al., 2003); above-surface winds;
or surface shear stress (Otto-Bliesner et al., 2006). Where sea ice replaces open water, each of these
diagnostics shows a different response (Sime et al., 2013). Sea ice affects surface roughness and
near surface stratification of the boundary layer, this can lead to quite different results for glacial
to interglacial changes in different wind diagnostics and shear stress (Figure 1). For this reason, we
concentrate on the above surface (850 hPa) winds, given that any model specific specification of
sea ice effects tends to have a lesser impact on this diagnostic (Sime et al., 2013). However, given
the importance of surface wind speed and shear stress for driving the Southern and global Ocean
circulation, some discussion of all of these three wind diagnostics is included in this study.

When calculating jet intensity and position for each diagnostic, we use a cubic spline interpolation
to quantify the jet maximum and determine its latitude. Jet shifts are defined here as PI to LGM
changes in the latitudinal position of the zonal mean maximum in the jet. Data is regridded to a
consistent 0.1° resolution before these calculations are performed. In addition to these zonal mean
diagnostics, we also assess individual ocean sectors results. In these cases, sectors are defined by
longitude ranges as follows: Atlantic sector (290° to 20°), Indian sector (20° to 150°) and Pacific
sector (150° to 290°). Jet diagnostics are calculated for the climatological maximum in the Southern
Hemisphere 850 hPa wind component; the climatological 1000 hPa westerly wind is used as an
indicator for surface wind. This diagnostic is used in lieu of the 10-metre surface westerly wind
speed 'uas' field, because 'uas' is not available for LGM simulations for two CMIP5 models. We
also calculate the zonal shear stress $\tau_U$, 'jet' position and intensity. Required variables ('ua' and
'tauu') were downloaded from the CMIP5 data archive between September and October in 2014.

All CMIP5 models show an equatorward bias in the present-day zonal mean surface jet position.
The ensemble of present day CMIP5 simulations show a mean equatorward bias of 3.3° (inter-
model standard deviation of $\pm$ 1.9°) in the position of the surface zonal mean jet (Bracegirdle et al.,
2013; Swart and Fyfe, 2012). A bias is still present, but is reduced, in atmospheric-only simulations.





This implies that sea surface temperature and sea ice errors, *i.e.* ocean-atmosphere coupling, tends
to generate wind jet biases. Simulations of Southern Ocean winds are similar for both standard
and high-top version models (Wilcox et al., 2012), which implies that inter-model differences in
stratospheric resolution and representation may not be critical. The equatorward jet biases are mainly
associated with the Indian and Pacific sectors.

## 2.2   The sea ice edge

Where available, sea ice concentration data was downloaded for the model simulations. The sea
ice edge was calculated using a mean annual sea ice concentration of 15%. For the few model
simulations sea ice concentration data was not available. In this case a best fit relationship between
sea surface temperature and sea ice edge, derived from the models where both output were available,
was used to estimate the sea ice edge (COSMOS-ASO and IPSL-CM5A-LR).

## 3   Results

### 3.1   Jet changes and state dependency

We focus in this study on the PI and LGM CMIP5-PMIP3 simulations. Table 2 indicates a wide
range of PI to LGM latitudinal jet shifts across the PMIP3 simulations, varying from $+ 2.0$ to $- 4.5$
° for the 850 hPa jet. The mean 850 hPa jet shift for the nine models is small: $- 0.2$ ° (inter-model
standard deviation of $\pm 2.1°$). The mean surface jet shift for the nine models is $- 0.9$ ° (inter-model
standard deviation of $\pm 1.6°$). The median shift for both 850 and 1000 hPa is 0°. Similar inter-model
variation appears in the jet intensity changes (Table 2).

Following the Bracegirdle et al. (2013) approach, we calculate state dependency for Southern
Ocean jet shifts and intensity changes across the various oceanic sectors, *i.e.* the dependence of PI
to LGM jet shifts with PI jet position. Feedbacks within the troposphere have been used to explain
state dependence in previous studies (*e.g.* Kidston and Gerber, 2010). We find that state dependency
can explain up to 56 % of the variance in PI to LGM jet shifts in the Atlantic (r = -0.75, N = 9, for
$\tau_U$), and 41 % in the Indian Ocean (r = -0.64, N = 9, for $\tau_U$). State dependency is much weaker in
the Pacific; here any influence is negligible. 850 hPa and 1000 hPa results are always very similar
(not shown). We find state dependence is stronger for the $\tau_U$ jet than the 850 hPa jet (Figure 2), due
largely to the MRI-CGCM3 850 hPa outlier. With the anomalous MRI-CGCM3 850 hPa wind result
removed from the calculation, we obtain similar results between 850 hPa and $\tau_U$. For the whole of
the Southern Ocean, the variance explained by state dependency is 38 % (r = -0.62, N = 9, for $\tau_U$).
Whilst these CMIP5-PMIP3 results bear similarities to the Bracegirdle et al. (2013) CMIP5-
RCP8.5 analysis of state dependence, they also show distinct differences. Over the Atlantic sector
Bracegirdle et al. (2013) find the correlation, calculated between present day and RCP8.5 CMIP5
output, is relatively weak (r = -0.39) compared with the correlations over the Indian sector (r =



-0.50) and Pacific sector (r = -0.91). Interestingly, Bracegirdle et al. (2013) also find that correla-

tion results over the Atlantic are conditional on omitting model MRI-CGCM3, which is again an influential outlier due to jumps in jet position.

For jet intensity Bracegirdle et al. (2013) find the state dependence is generally weaker than for position; we find a similar result here. The state dependency in intensity change can explain only 25 % (r = -0.50, N = 9, for $\tau_U$) of the PI to LGM change.

This analysis indicates that, whilst state dependence plays a role in determining deglacial-warming jet shifts and intensity changes, overall the influence of state dependency alone is much weaker compared with future-warming climate change scenarios. This implies that other factors must also be important in determining the LGM wind changes; we now look at the factors which are most likely to drive these changes.

## 3.2 The impact of sea ice

The most recent compilation of LGM sea surface temperature data is the MARGO dataset (MARGO Project Members, 2009). Although the coverage of MARGO data is good in tropical regions, it is sparse poleward of 40°S (MARGO Project Members, 2009; Sime et al., 2013). However, Gersonde et al. (2005) provide LGM sea surface temperature and sea ice data from 122 Southern Ocean sed-

iment core sites. This data suggests that LGM sea ice extended in the Atlantic and Indian sector to close to 47°S , and in the Pacific sector as far north as 57°S; a PI to LGM equatorward expansion of some 7 to 10° in latitude. This is a large change, particularly compared with the sea ice changes which occur during most future-warming scenario CMIP5 simulations.

All CMIP5-PMIP3 models for which we can retrieve sea ice output show an LGM expansion of

sea ice in the Southern Hemisphere (Table 3). There is considerable variability between the models. Expansions range between 2.1 and 7.0 ° (Figure 4, Table 3). Only two models, CCSM4 and MRI-CGCM3 (with Gersonde et al. (2005) data agreements of 87% and 88%, see Appendix 1), appear to yield an accurate simulation of LGM sea ice extent (Table 3), and some of the largest equatorward expansions of sea ice at 5.6 and 7.0 °, respectively.

Changes in sea ice extent are associated with relatively strong surface heat flux anomalies, which can be as large as 100 W m$^{-2}$ (Alexander et al., 2004). A strong non-linearity of wind response can thus be generated, dependent on the location of the resultant changes in meridional temperature gradients in the atmosphere. For example, surface cooling due to an expansion of sea ice causes an anomalous increase in the meridional temperature gradient adjacent to the newly ice-covered ocean.

If this increased gradient lies immediately poleward of the jet and its associated baroclinic zone, it can be more effective at influencing developing baroclinic waves and the latitude of the jet. Support for this idea is also found in the results of Chen et al. (2010) and Brayshaw et al. (2008), where changes in surface heat fluxes have the largest impact when they are approximately co-located with the maximum in the meridional temperature gradient. If we look at changes in CMIP5-PMIP3 sea



ice along with westerly winds and meridional temperature gradients throughout the atmosphere, we see evidence of this effect (Figures 5 and 6).

When looking at all models, there are some commonalities in the meridional structure in wind and temperature gradient changes. Between 0 and 200 hPa and poleward of 50°S the temperature gradient increases, resulting in an increase in westerly wind speed, U. Equatorward of 50°S the tem-

perature gradient tends to decrease, with the strongest decrease between 50 and 300 hPa. Upper and mid tropospheric U also decreases in all models equatorward of around 50°S. Below 400 hPa there tends to be an increase in the meridional temperature gradient poleward of around 40°S, however there is considerable inter-model variability in the details of the temperature gradient changes and in associated wind changes.

As the above implies we find that the key differences, in westerly winds and meridional temperature gradients changes, are a function of state dependence and sea ice. Indeed, based on state dependence (PI jet position) and sea ice changes, models can be roughly classed into four groups. CCSM4 and MRI-CGCM3, in the first group (Figures 5ab and 6ab), both simulate large expansions in sea ice (of 5.6 and 7°, respectively) and feature the most southerly positioned jets (at 51°S ±

1.1°). This jet position tends to leave U sensitive to expansions in sea ice. The large increases in the meridional temperature gradient, especially around 55°S from 1000 to about 650 hPa, thus tally with the increases in U around these latitudes, and also results in a large PI to LGM poleward shift in the jet in MRI-CGCM3.

In the second group, whilst GISS-ER-R and CNRM-CM5 (Figures 5cd and 6cd) have PI jets

which are positioned relatively far to the south (at 48°S ± 0.3°) both feature rather small LGM expansions in sea ice (of around 2°). The resultant small polar atmospheric cooling causes little change in the meridional temperature gradient. U tends to weaken over the Southern Ocean latitudes, likely due to the overall atmospheric cooling. These two models show a slight equatorward shift in their jets between the PI and the LGM.

In the third group, COSMOS-ASO and MPI-ESM-P (Figures 5ef and 6ef) have PI jets positioned at 47°S ± 0.6°, and feature quite large LGM expansions in sea ice (of 5 and 6 °, respectively). The position of these jets makes these models less sensitive to the LGM expansion of sea ice: they show a slight weakening of U and no jet shifts. It seems that the climatological storm track and its associated baroclinic zone is not significantly affected by these sea ice increases, and associated

meridional temperature gradient changes, because they happen far poleward of the baroclinic jet zone.

In the last group, FGOALS-G2, IPSL-CM5A-LR, and MIROC-ESM (Figures 5ghi and 6ghi) all have very northerly positioned PI jets (at 43°S ± 0.5°), they also all show a rather small (less than 3°) LGM increase in sea ice extent. The jets in these models thus seem to be responding to

influences other than sea ice: possibly tropical changes, or sea surface temperature changes nearer 43° have more impact. Chavaillaz et al. (2013) find a quasi-linear relationship between the jet shifts



and tropical temperature changes in the atmosphere, where polar temperatures are held constant, suggesting that tropical changes may be a stronger influence on these models.

### 3.2.1 The relationship between sea ice extent and jet position

In simulations with more poleward (*i.e.* accurately) positioned PI jets, the examination above of jet and sea ice changes suggests that PI to LGM wind changes are strongly related to sea ice extent. Figure 7a shows that the PI jet position is inversely related to sea ice extent, in models with the most accurately positioned PI jets. We find that an equatorwards sea ice edge correlates with a poleward jet position (r = -0.95 for the PI, and r = -0.91 for the LGM). Whilst correlations are strongest for

850 hPa winds, similar results are obtained using 1000 hPa and $\tau_U$ (r < -0.80).

  In terms of PI to LGM jet shifts, if we apply a linear least-squares fit, we find that a 1° difference in the sea ice edge suggests a -0.85° shift in the 850 hPa jet (r = -0.80; N=5). These results are heavily influenced, but are not entirely dependent on, the MRI-CGCM3 model. This model features the largest 7° expansion of the sea ice and a large 4.5°S poleward shift in the 850 hPa jet (Figure 7;

Table 3). Without this model result included in the calculation, a 1° difference in the sea ice edge still suggests a -0.43° shift in the 850 hPa jet (r = -0.96; N = 4). However, as the section above indicates, this relationship only applies to models which have jets which are relatively accurately positioned *i.e.* those which are sensitive to the impact of sea ice changes; if the model has a jet which sits equatorward of 47°S then the relationship breaks down. This relationship also fits with

the study of Kidston et al. (2011) who found that the jet shifts significantly poleward when the sea ice extent is substantially increased. However the jet exhibits little response for small changes, and particularly little response if the sea ice edge is far from the jet, for example during the summer, when the sea ice edge is far from the jet maxima. The cause of the asymmetry in the atmospheric response relates to the extent to which sea ice changes affect meridional temperature gradients in

the near-surface baroclinic zone. Together these results suggest that the expansion of sea ice during the LGM is critically important to jet shifts, but only if the PI jet is close enough, *i.e.* is positioned accurately enough, relative to the PI sea ice edge.

  In addition, the offset of the fitted line in Figure 7b suggests that, without any expansion in sea ice, the jet might tend to shift towards the equator; by around 4° during the LGM. From the zero-cross

of the line, we tentatively suggest that around 5° of sea ice expansion is necessary to counteract this tendency. Given that the Gersonde et al. (2005) data supports a latitudinal expansion of more than 5°, this result does suggest that a slight poleward shift (and intensification) is likely to have been a feature of the LGM jet.



### 3.2.2 Sea surface temperatures changes

If we also fit a linear model to jet shifts against sea surface temperature changes in the marginal
sea ice zone we find a weak positive relationship between sea surface temperature and 850 hPa jet
position.

   Given the strong relationship between Southern Ocean surface temperature and sea ice, it is dif-
ficult to separately assess any influences of sea ice and sea surface temperature on Southern Ocean
winds. However, Sime et al. (2013) conducted sensitivity experiments, using an atmospheric-only
GCM, in order to attempt to elucidate these relationships. As the analysis above suggests, Sime et al.
(2013) showed that cooling the Southern Ocean, near the sea ice edge, around 55°S, and extending
the sea ice promote the same response in the 850 hPa winds.

   Here, with CMIP5-PMIP3 results, we find that for the five models with PI jets which are positioned
poleward of 47°S an average temperature change of $-1$ K (over the Gersonde et al. (2005) data
network locations) results in a 3.0° poleward shift in the 850 hPa jet (r = 0.83; n=5; Figure 7c).
Sime et al. (2013) also found that cooling near the edge of the LGM Southern Ocean sea ice and
extended sea ice coverage caused a wind intensification which is largest between 56-58°S. This
drives the small poleward shift in the location of the winds maximum. Here, CMIP5 models with
accurately positioned PI jets, show a similar result.

### 4    Summary and conclusions

We have analysed the CMIP5-PMIP3 LGM and PI simulations for Southern Ocean region wind
changes, and examined the impacts of sea surface changes and state dependency. Nine fully inde-
pendent CMIP5-PMIP3 model simulations were included in the analysis. We find a wide range of
PI to LGM latitudinal shifts in the jet across the PMIP3 simulations, varying from $+ 2.0$ to $- 4.5$ °
for the 850 hPa jet, but the mean 850 hPa jet shift for the nine models is small: $- 0.21$ °. The de-
pendence of PI to LGM jet shifts on PI jet position, referred to here as state dependency (following
Bracegirdle et al., 2013), explains up to 56 % of the variance in PI to LGM jet shifts in the Atlantic
sector of the Southern Ocean, for $\tau_U$, and 41 % in the Indian sector. Since state dependence plays a
weaker role in determining jet shifts and strength changes for the deglacial-warming, compared to
future-warming scenarios, this implies that other factors are important in determining the LGM wind
changes. Changes in surface heat fluxes, due to sea ice changes, can have very large impact on the jet,
particularly when they are located close to the position of the jet, which is itself usually co-located
with the maximum in the meridional temperature gradient. Given that LGM sea ice extended in the
Atlantic and Indian sector close to 47°S , and in the Pacific sector as far north as 57°S (Gersonde
et al., 2005), the key differences in jet shifts seem to be a dual function of state dependence and sea
ice change.





All CMIP5-PMIP3 models show an LGM expansion of sea ice in the Southern Hemisphere, but there is considerable inter-model variability in the size of the expansion, which ranges from 2.1 and

7.0 °. State dependence (PI jet position) and sea ice changes together control jet shift behaviours. Only two models, CCSM4 and MRI-CGCM3, both simulate realistically large expansions in sea ice and simulate PI jets which are south of 50°S. These models show an increase in U around 55°S, and a large PI to LGM poleward shift in the jet in MRI-CGCM3. For models which have jets that are positioned relatively far to the south (at 48°S ± 0.3°) but which do not correctly simulate the

observed expansion in sea ice, the resultant small polar atmospheric cooling causes little change in the meridional temperature gradient. In this case, U simply tends to weaken over the Southern Ocean latitudes. Models which simulate a large increase in sea ice extent, but which have jets positioned too far towards the equator are not sensitive to the LGM expansion of sea ice: they show a slight weakening of U, but no jet shifts. The jet is not significantly affected by these sea ice increases, and

associated meridional temperature gradient changes, because they happen too far poleward of the baroclinic jet zone. This also fits with the study of Kidston et al. (2011) who found that whilst the jet will shift significantly poleward when the sea ice extent is substantially increased, there is little response if the sea ice edge is far from the jet.

We can generalise the relationship between sea ice extent and jet position. In models with ac-

curately positioned PI jets; a 1° difference in the sea ice edge tends to be associated with a -0.85° shift in the 850 hPa jet. However, without any expansion in sea ice, it seems that the jet would shift towards the equator, by around 4° during the LGM. Thus we tentatively conclude that around 5 ° of sea ice expansion is necessary to hold the jet in its PI position. Given that the Gersonde et al. (2005) data supports a northward expansion of more than 5 °, this result does suggest that a slight poleward

shift (and intensification) is likely to have been a feature of the LGM jet. This fits with the findings of Sime et al. (2013), who found that cooling near the edge of the LGM Southern Ocean sea ice caused a wind intensification which is largest between 56-58°S. But we emphasise that these results only apply to CMIP5 models which have jets which are relatively accurately positioned *i.e.* those which are sensitive to the impact of sea ice changes; if the model has a jet which sits equatorward of

about 47°S then the relationship breaks down.

Finally, one perhaps overlooked feature of the CMIP5 present day simulations is the proliferation of simulations which feature surface jets 5° equatorward of the current position. Many CMIP5 simulations also feature jets which can be 20% too weak (Bracegirdle et al., 2013). CMIP5 models can thus simulate jets which are considerably farther equatorward and weaker than they are in

our current climate. Further work could investigate whether these jet positions could themselves be taken as evidence for the physical feasibility of the large LGM jet shift hypothesis. The output from these simulations could perhaps also be compared with ongoing work on the development of reliable paleo-wind and sea ice reconstructions. Pending this however, we conclude from our analysis of CMIP5-PMIP3 output that the LGM Southern Ocean extended sea ice coverage was most likely



responsible for a small wind intensification, which was largest around 55-58°S. Without the effect of sea ice and associated sea surface cooling, models simulate polewards shifted westerlies in warming climates and equatorward shifted westerlies in colder climates. However, the impact of LGM sea ice counters and reverses the equatorward trend in cooler climates so that the LGM winds were more likely to have also been shifted slightly poleward.

## 5   Appendix 1: Observational data and calculating model-data agreements

The most recent compilation of LGM sea surface temperature data is the MARGO dataset (MARGO Project Members, 2009). The coverage of MARGO data is particularly good in tropical regions, but is sparse poleward of 40°S (MARGO Project Members, 2009; Sime et al., 2013). However Gersonde et al. (2005) provide LGM sea surface temperature and sea ice data from 122 Southern
Ocean sediment core sites. This data suggests that the maximum winter sea ice extent extended in the Atlantic and Indian sector close to 47°S , and in the Pacific sector as far north as 57°S, an equatorward displacement of 7 to 10° in latitude across the Southern Ocean (Figure 4).

Sime et al. (2013) suggested using simple model-data evaluation metrics based on simple percentage statistics. Here we use a similar approach, and assess sea ice model-data agreement by classing
simulated sea ice as present or absent, rather than using concentration values. Simulation results are bi-linearly interpolated to the observation site prior to the model-data assessment, analogous to the exact-metric defined by Sime et al. (2013).

*Acknowledgements.* The work was funded by NERC grants NE.K004514.1 and NE/J004804/1, and also forms part of the British Antarctic Survey Polar Science for Planet Earth Programme. We acknowledge the support
of the ARCHER UK National Supercomputing Service (http://www.archer.ac.uk), and the World Climate Research Programme's Working Group on Coupled Modelling, which is responsible for CMIP, and we thank the all the climate modeling groups for producing and making available their model output.





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



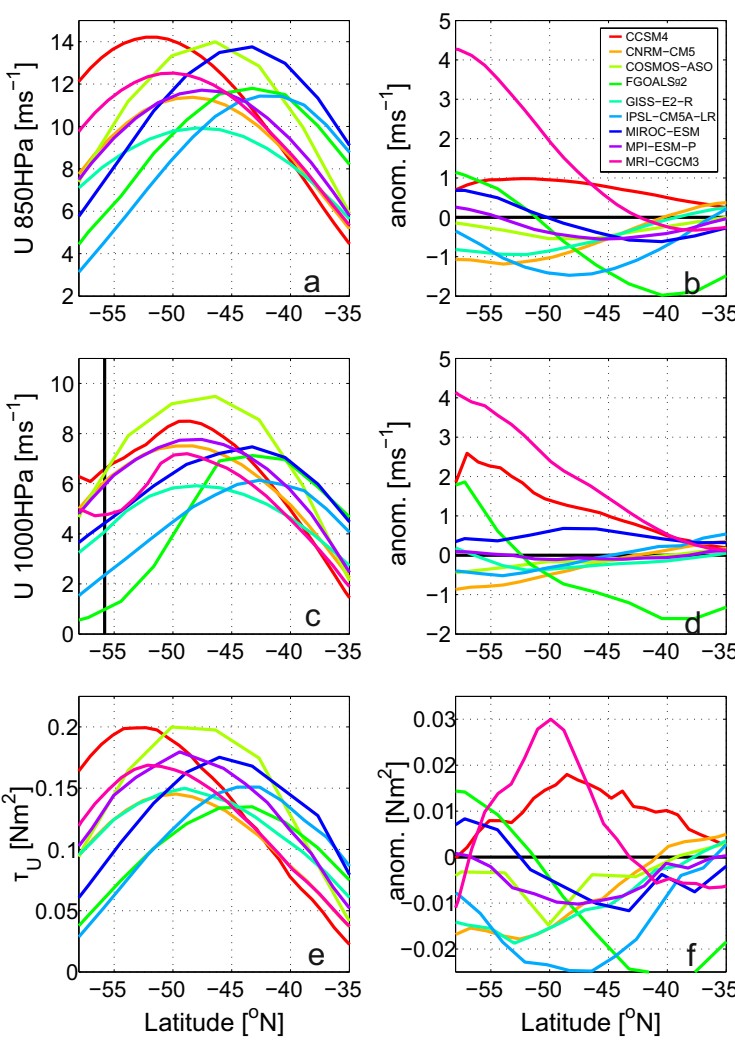

**Figure 1.** Zonal mean Southern Ocean winds. (a) PI wind speed U at 850 hPa, (b) and the LGM − PI anomaly, (c) PI wind speed U at 1000 hPa, (d) and the 1000 hPa LGM − PI anomaly, (e) PI surface shear stress $\tau_U$, (f) and the $\tau_U$ LGM − PI anomaly. Colours as shown in the legend on panel (b) denote the individual models.





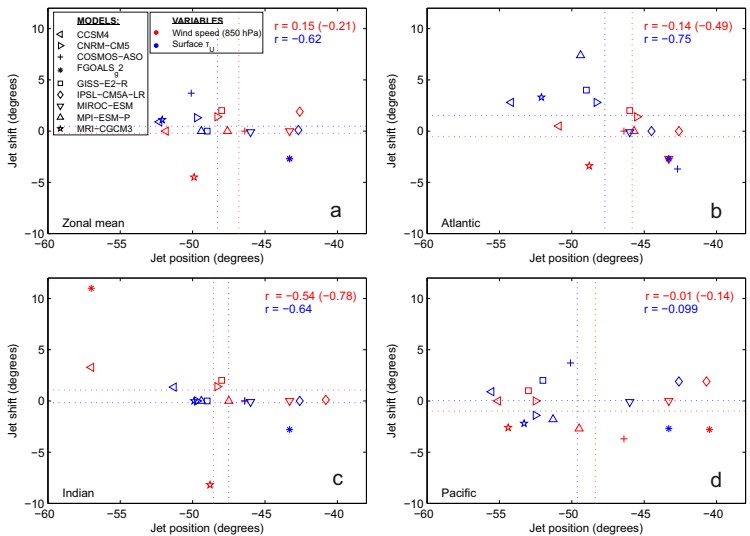

**Figure 2.** The state dependence of jet latitudinal shifts. (a) Scatter plot of PI annual mean Southern Ocean jet position versus LGM minus PI change in the CMIP5-PMIP3 models. (b), (c), and (d) show the same but for individual ocean sectors: Atlantic, Indian, and Pacific, as marked. The Atlantic sector is defined as 290° to 20°, the Indian as 20° to 150°, and the Pacific as 150° to 290°. Individual symbols indicate individual models, as marked in the legend in panel (a). The jet is defined using the zonal 850 hPa wind speed (red), and surface shear stress (blue). Dashed lines indicate the model mean position and model mean shift for each panel. Specified r values indicate the correlation coefficient; bracketed 850 hPa r values are calculated excluding the MRI-CGCM3 model. Correlation coefficients and relationships using 1000 hPa 'surface wind speeds' are almost identical to those using 850 hPa.





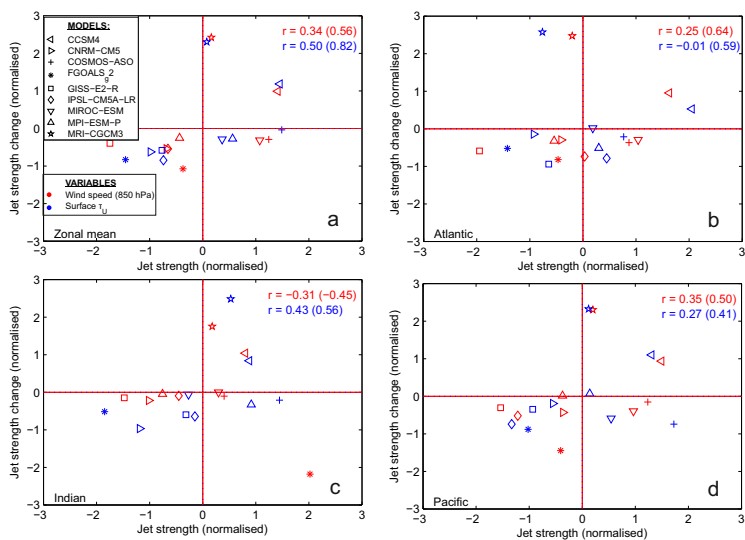

**Figure 3.** The state dependence of jet intensity (wind speed) changes. (a) Scatter plot of PI annual mean Southern Ocean jet strength versus LGM minus PI change in the CMIP5-PMIP3 models. Zonal mean (a), Atlantic sector (b),Indian sector (c), and Pacific sector (d). Results are normalised by subtracting the mean of all models, and dividing through by the standard deviation of all models. Caption and results as Figure 2.





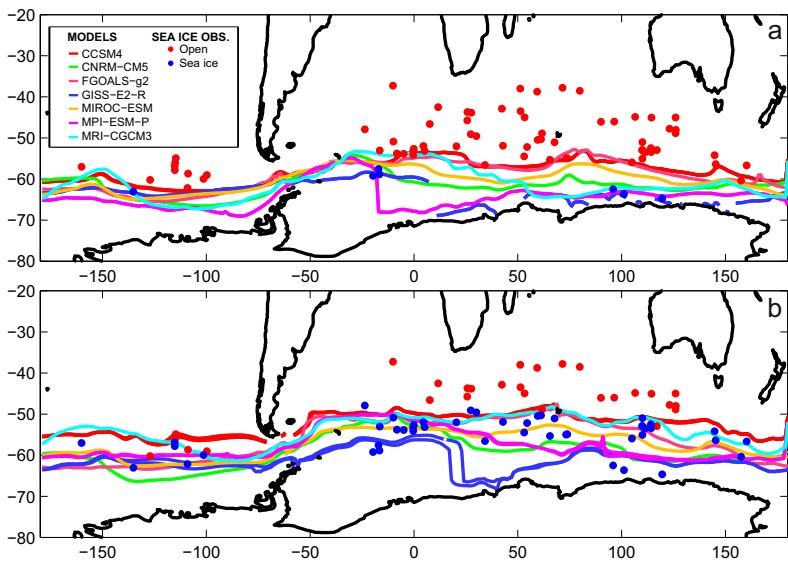

**Figure 4.** Sea ice from models and the Gersonde et al. (2005) observations for (a) the PI and (b) the LGM. Coloured dots show open water (red) and inferred sea ice (blue). The differing coloured lines show the annual mean 15% sea ice extent for individual models. See legend for colours.





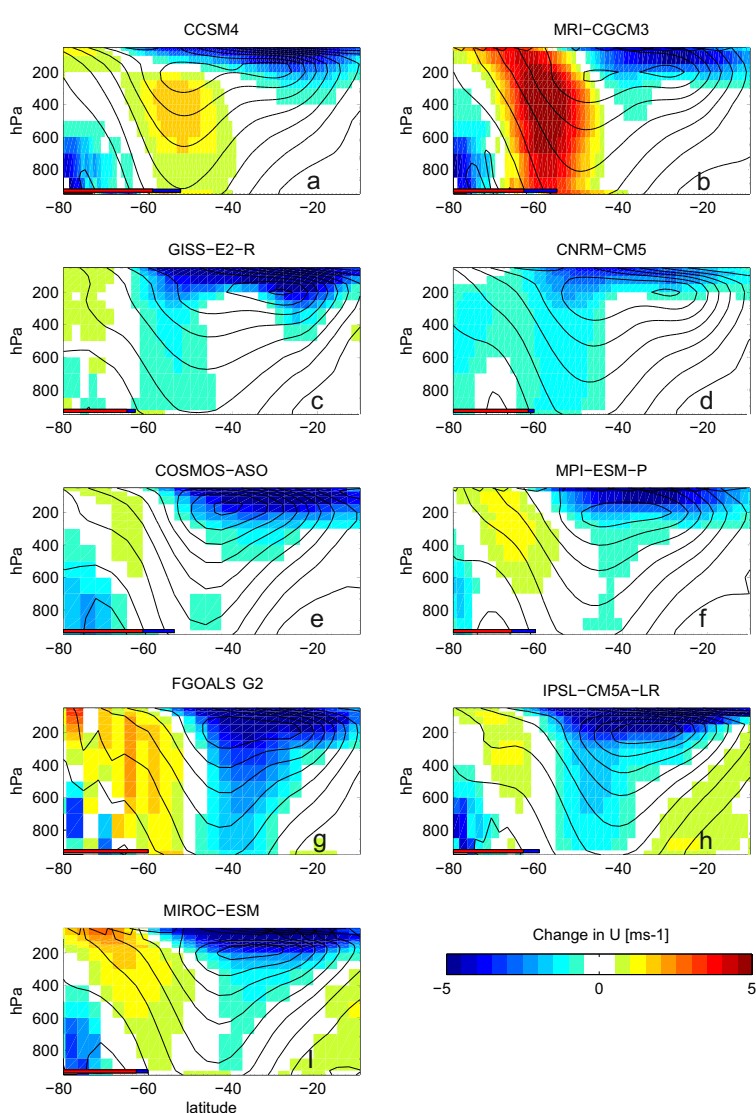

**Figure 5.** Change is the zonal mean wind velocity component in the westerly direction U (LGM − PI) through-out the atmosphere (shaded). Black contours show mean U for the PI. Red and blue bars at the bottom left of each panel indicate the extent of the zonal mean sea ice for the PI and LGM, respectively. Individual panels are labeled to indicate individual models.




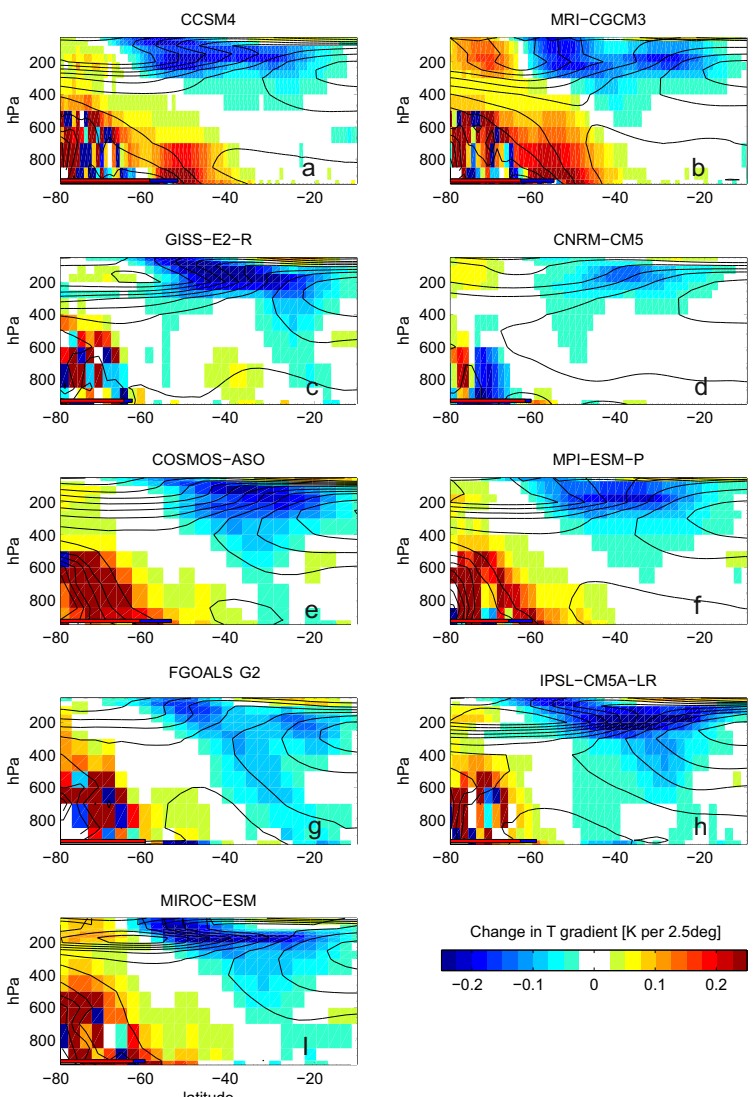

**Figure 6.** Change is zonal mean temperature gradient (LGM − PI) throughout the atmosphere (shaded). Black contours show the mean temperature gradient for the PI. All temperature gradients are meridional. Red and blue bars at the bottom left of each panel indicate the extent of the zonal mean sea ice for the Pi and LGM, respectively. Individual panels are labeled to indicate individual models.



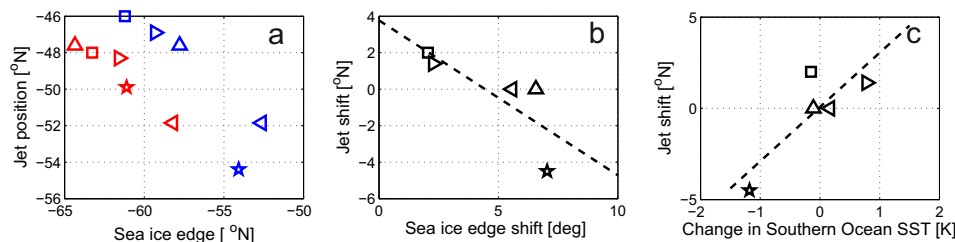

**Figure 7.** The relationship between sea ice, SST, and the position of the 850 hPa jet. Symbols represent individual models as shown in Figure 2. Red indicates the PI; blue indicates the LGM; and black indicates changes between the PI and the LGM (LGM − PI). (a) The jet position against sea ice extent (15%); (b) jet shift against sea ice extent change; and (c) jet shift against Southern Ocean SST change. We use models where the PI jet position is poleward of 47 °S, and results are interpolated to the position of the Gersonde et al. (2005) observations before each calculation.



**Table 1.** List of all CMIP5-PMIP3 simulations used in this study. The individual simulations (r < N > i < M > p < L > ) formatted as shown below (e.g. 'r3i1p21' with r for 'realization', i for 'initialization method indicator' and p for 'perturbed physics') distinguishes among closely related simulations by a single model.

| Model | Institute | Grid size | | Simulations | |
| --- | --- | --- | --- | --- | --- |
| | | Lat. | Long. | Control (PI) | LGM |
| CCSM4 | National Center for Atmospheric Research, U.S. Dept. of Energy/NSF | 192 | 288 | r1i1p1,r2i1p1 | |
| CNRM-CM5 | Centre National de Recherches Météorologiques/Centre Européen de Recherche et Formation Avancée en Calcul Scientifique, France | 128 | 256 | r1i1p1 (CNRM-CM5) | r1i1p1 (CNRM-CM5) |
| COSMOS-ASO | Max Planck Institute for Meteorology, Hamburg, Germany | 48 | 96 | r1i1p1 (COSMOS-ASO) | r1i1p1 (COSMOS-ASO) |
| FGOALS-g2 | Institute of Atmospheric Physics, Chinese Academy of Sciences and CESS, Tsinghua University, China | 60 | 128 | r1i1p1 (FGOALS-g2) | r1i1p1 (FGOALS-g2) |
| GISS-E2-R | NASA Goddard Institute for Space Studies, | 90 | 144 | r1i1p142 (GISS-E2-R-p150) r1i1p142 (GISS-E2-R-p151) | r1i1p150 (GISS-E2-R-p150) r1i1p151 (GISS-E2-R-p151) |
| IPSL-CM5A-LR | Institut Pierre-Simon Laplace, France | 96 | 96 | r1i1p1 (IPSL-CM5A-LR) | r1i1p1 (IPSL-CM5A-LR) |
| MIROC-ESM | Japan Agency for Marine-Earth Science and Technology, Atmosphere and Ocean Research Institute (The University of Tokyo), and National Institute for Environmental Studies, Japan | 64 | 128 | r1i1p1 (MIROC-ESM) | r1i1p1 (MIROC-ESM) |
| MPI-ESM-P | Max Planck Institute for Meteorology, Hamburg, Germany | 96 | 192 | r1i1p1 (MPI-ESM-P-p1) r1i1p1 (MPI-ESM-P-p2) | r1i1p1 (MPI-ESM-P-p1) r1i1p2 (MPI-ESM-P-p1) |
| MRI-CGCM3 | Meteorological Research Institute, Tsukuba, Japan | 160 | 320 | r1i1p1 (MRI-CGCM3) | r1i1p1 (MRI-CGCM3) |



**Table 2.** The Southern Ocean Westerly Winds jet position, jet strength and PI to LGM changes in jet position and strength.

| Model | Location PI jet [°N] | | | Jet shift [°] | | | Maximum in PI Jet | | | Jet strength change | | |
|---|---|---|---|---|---|---|---|---|---|---|---|---|
| | 850 hPa [°N] | 1000 hPa [°N] | $\tau_U$ [°N] | 850 hPa [°] | 1000 hPa [°] | $\tau_U$ [°] | 850 hPa [ms$^{-1}$] | 1000 hPa [ms$^{-1}$] | $\tau_U$ [Nm$^{-2}$] | 850 hPa [ms$^{-1}$] | 1000 hPa [ms$^{-1}$] | $\tau_U$ [Nm$^{-2}$] |
| CCSM4 | -51.8 | -49.0 | -52.3 | 0.0 | -1.4 | 0.9 | 14.2 | 8.53 | 0.199 | 0.99 | 1.50 | 0.011 |
| CNRM-CM5 | -48.3 | -48.4 | -49.7 | 1.4 | 0.1 | 1.3 | 11.4 | 7.51 | 0.145 | -0.84 | -0.38 | -0.015 |
| COSMOS-ASO | -46.4 | -46.4 | -50.1 | 0.0 | 0.0 | 3.7 | 14.0 | 9.49 | 0.20 | -0.54 | -0.18 | -0.006 |
| FGOALS g2 | -43.3 | -43.3 | -43.3 | -2.7 | -2.7 | -2.7 | 11.8 | 7.12 | 0.135 | -1.48 | -1.21 | -0.018 |
| GISS-E2-R | -48.0 | -48.0 | -49.0 | 2.0 | 0.0 | 0.0 | 9.9 | 5.92 | 0.150 | -0.67 | -0.31 | -0.014 |
| IPSL-CM5A-LR | -42.6 | -42.6 | -42.7 | 1.9 | 0.0 | 0.1 | 11.4 | 6.14 | 0.151 | -0.84 | 0.12 | -0.018 |
| MIROC-ESM | -43.3 | -43.3 | -46 | 0.0 | 0.0 | -0.1 | 13.8 | 7.47 | 0.175 | -0.58 | 0.54 | -0.010 |
| MPI-ESM-P | -47.6 | -47.6 | -49.4 | 0.0 | 0.0 | 0.0 | 11.7 | 7.77 | 0.180 | -0.50 | -0.07 | -0.010 |
| MRI-CGCM3 | -49.9 | -48.8 | -52.1 | -4.5 | -4.5 | 1.1 | 12.5 | 7.20 | 0.169 | 2.70 | 3.03 | 0.027 |
| MEAN | -46.8 | -46.4 | -48.3 | -0.21 | -0.94 | 0.48 | 12.3 | 7.46 | 0.167 | -0.196 | 0.34 | -0.006 |



**Table 3.** The model-data agreement from the sea ice edge latitude; the mean zonal sea ice edge latitude; and the PI to LGM jet shift. Simulation results are bi-linearly interpolated to each Gersonde et al. (2005) observation site. Model-data agreement is then calculated by classing simulated sea ice as present or absent, analogous to the exact-metric defined by Sime et al. (2013). A simple agreement percentage metric is then calculated using the equivalent Gersonde et al. (2005) sea ice (present or absent) observations.

| Model | Sea ice agreement | | Sea ice edge position | | Sea ice shift |
| --- | --- | --- | --- | --- | --- |
| | PI | LGM | PI | LGM | LGM-PI |
| | [%] | [%] | [$^o$N] | [$^o$N] | [$^o$] |
| CCSM4 | 94 | 87 | -58.2 | -52.6 | 5.57 |
| CNRM-CM5 | 98 | 56 | -61.6 | -59.3 | 2.27 |
| COSMOS-ASO | NaN | NaN | NaN | NaN | NaN |
| FGOALS-g2 | 85 | 37 | -59.2 | -56.4 | 2.78 |
| GISS-E2-R | 85 | 37 | -63.3 | -61.2 | 2.05 |
| IPSL-CM5A-LR | NaN | NaN | NaN | NaN | NaN |
| MIROC-ESM | 85 | 37 | -60.3 | -57.4 | 2.91 |
| MPI-ESM-P | 87 | 59 | -64.3 | -57.8 | 6.57 |
| MRI-CGCM3 | 92 | 88 | -61.1 | -54.1 | 7.04 |