# Peer review of "Sea ice led to poleward-shifted winds at the Last Glacial Maximum: the influence of state dependency on CMIP5 and PMIP3 models"

_Climate of the Past, 2016_

## Referee Comment (RC1) · Anonymous Referee #1 · 23 May 2016

General Comments This manuscript addresses the issue of how the southern hemisphere westerly winds changed during the last glacial maximum. Changes in the winds have been invoked as a significant player in the glacial-interglacial $CO_2$ variations (the o called "wind-hypothesis"), hence there has been in recent years a number of papers trying to gain insight on this issue from a modelling perspective. A conclusive support or disapproval of the "wind hypothesis" has remained elusive. This paper uses the same existing simulations reported in at least 3 other papers (Chavaillaz et al, 2013, Rojas, 2013 and Liu et al, 2015), in this sense there is a feeling that the results presented are not very new. The main hypothesis for explaining the variety of wind responses to LGM conditions is the sea-ice response of the model and the

state dependency. The sea-ice hypothesis has also been invoked in other papers, but the complete analysis of the relationship with the presented day simulation of the winds is novel. In this respect the paper is a contribution towards settling this long-standing issue. However there are at least two important papers that have not been cited: Liu et al 2015: The de‐correlation of westerly winds and westerly‐wind stress over the Southern Ocean during the Last Glacial Maximum. Journal of Climate At the end we are interested in understanding if changes in winds can be responsible for part of the glacial-interglacial atmospheric CO2 variations, by helping to outgass CO2 from the southern ocean. This aspect is addressed in Liu et al 2015. And in this paper this issue is left at a secondary plane. Also, given that sea-ice is diagnosed as a critical component of the LGM response of the system in the southern hemisphere, authors should discuss: Roche, D. M., Crosta, X., & Renssen, H. (2012). Evaluating Southern Ocean sea-ice for the Last Glacial Maximum and pre-industrial climates: PMIP-2 models and data evidence. Quaternary Science Reviews, 56(C), 99–106. http://doi.org/10.1016/j.quascirev.2012.09.020

This paper is PMIP2, so it would be interesting to include in the analysis presneted in this paper PMIP2 simulations as well (Rojas 2013, shows the zonal mean winds in PMIP2 and PMIP3).

Although the title refers to the conclusion that sea-ice must have been responsible for shifting the SHW poleward during the LGM, the discussion in the paper is too much centered on the modelling errors and how those errors have therefore hindered to date to use models to prove or disprove the "wind-hypothesis". I would re-emphasis the final objective which is to know if the winds where weaker, stronger or shifted during the LGM compared to present day, or pre-industrial climate. Paper requires revisions.

Specific Comments

Line 66: is PI defined? Line 69: include reference to Liu et al 2015 Line 113: include: "..global Ocean circulation and hence CO2 exchange.." Line 135: Why do you

only analysis annual means, when literature seems to indicate that summer sea-ice is important?? Lines 153-155: can you put those values in a table? Line 178: Does the Geresonde paper indicate summer/winter differences? Line 203: change 0 hPa to top level? Line 217-218: "...poleward shift in the jet in both models, especially in MRI-CGCM3". Line 261: but can only be captured if the PI jet position is accurately simulated. Line 262: include discussion of real world! Lines 265-268: discuss implication for CO2.

In section 4: Can you discuss more on the implication of your finding on CO2 variations? How reliable is the sea-ice reconstruction?

---

## Referee Comment (RC2) · Anonymous Referee #2 · 24 May 2016

Sime et al. use nine different CMIP5-PMIP3 preindustrial and LGM pairs to investigate the changes in Southern Hemisphere jet position and intensity and how would these changes related to the preconditioned jet location (state dependency) and sea ice expansion. This study is established on previous work by the author (i.e. Sime et al., 2013) and many other works by Kidston and Gerber (2010), Chavaillaz et al. (2013), Bracegirdle et al. (2013)….etc. and further suggest that sea ice being an important factor for the deglacial changes in Southern Hemisphere jet. It is, however, a bit weak on the discussion on how this study agree/disagree from previous studies and basin scale detail.

Specific Comments: 1. Do we confident on the actual condition of LGM SH jet condition

regardless of mutil-model mean suggests on no significant changes? If positive, then the conclusion that sea ice expansion holding jet in its present day position would valid otherwise the explanation is only for modeling perspective.

2. In line 117, "data is regridded to a consistent 0.1° resolution before these calculation are performed." I understand this is to separate the jet latitude between runs but wonder if it is legitimate to so. This is a one to twenty scaling after all as most of the model simulations here has a spatial resolution of 2.5°. One generally would not interpolate a T42 simulation to T106. By reading Table 2, it should do the job by interpolating data down to 0.5°. Or it would be nice to show the conclusion is not resolution (aka interpolation) dependent.

3. Section 3.1 describes the state dependency in PI-LGM changes. Can the author comment on why these results being quite different from Bracegirdle et al., (2013)? Bracegirdle et al. (2013) suggest strong dependency of jet over Pacific basin in warming scenario from PI to future condition while this study suggest much weaker state dependency in Pacific. Would this related to the different simulated sea ice and temperature conditions between PI-LGM and RCP-PI? This might further support the argument in Section 3.2.1 and 3.2.2.

4. Continue from previous comment, section 3.2 discuss the impact of sea ice. Is it possible to calculate the percentage of variation explained by state dependency and sea ice separately? In other word, which factor represents a stronger control over PI-LGM jet variability?

5. In reading Figure 5 and 6, it shows a non-proportional changes between temperature gradient structure and U wind changes. Can the author comment on this? For example, COSMO and MPI-ESM both show substantial changes in temperature gradient while MPI-ESM simulate none changes in zonal wind.

6. Suggestion: The authors mention in the manuscript that the changes in sea ice might be important in determining LGM SH jet changes. It can be verify and support by

comparing simulations with different sea ice extent, say LGM-PI-RCPs from extensive sea ice to sea ice free.

7. Very minor: in line 171 and line 300, as far "north" as 57°S, is this a typo of "south" relative to 47°S

---

## Referee Comment (RC3) · Anonymous Referee #3 · 26 Jun 2016

Review of "Sea ice led to poleward-shifted winds at the Last Glacial Maximum: the influence of state dependency on CMIP5 and PMIP3 models" by Louise Sime et al

General comments:

This manuscript investigates the Last Glacial Maximum (LGM) southern westerlies, a subject of high interest due to their possible control of the atmospheric CO2, whose LGM level is still not fully understood, but also due to their importance in atmosphere-ocean exchanges and ocean circulation. The southern westerlies are predicted to shift poleward under climate warming, and would therefore be expected to shift equatorward during the cooler-than-present LGM. Previous work has shown that this is not the case. This manuscript further analyses the possible implication of sea ice changes in LGM

southern jet changes. It is based on the PMIP3-CMIP5 data, which is the most recent data available from several state-of-the-art models run with the same boundary conditions. For sea-ice, this data is compared to the reconstructed data set of Gersonde et al (2005). The authors show that models which are sufficiently accurate for the present climate in their representation of the jet stream, i.e. for which the position of the jet stream maximum is not too much biased northward, and simulate a realistic sea-ice extension from present to LGM, can show a poleward shift of the jet stream for the LGM compared to present. Although the southern westerlies had been investigated in previous publications, this work goes one step further by 1/ clearly showing the impact on present day biases on the LGM – present changes and 2/ showing the role of sea-ice for those models which simulate realistic westerlies for the present situation. The manuscript is clearly written and illustrated, I am therefore in favour for its publication in Climate of the Past, after the corrections and improvements listed below.

Suggested improvements:

1. Since there are not so many models which have provided the complete data set for this analysis (in particular sea ice concentration) the authors may want to encourage the corresponding modelling groups to submit their data to the CMIP5-PMIP3 data base. This might increase the numbers used to infer the relationship between the jet shift and sea-ice edge or SSTs on Fig. 7.

2. Another way to increase the number of models, by using PMIP2 data, could also help, but these models being of an older generation might show even stronger biases?

3. Finally, would the study of the yearly fields help and quantify the uncertainty of the results shown on Fig. 7 (regression between jet position and sea ice edge)? e.g. by using bootstrapping to add information on the variability of the results?

4. One aspect of the manuscript I am very confused with is the seasonal/annual averaging of the data. On the one hand, section 2.1 specifies that "jet diagnostics are calculated for the climatological maximum in the Southern Hemisphere 850 hPa wind

component" but the season (or month?) is not specified. On the other hand, Figures 2, 3 and 4 show annual means of winds and sea ice. Given the large seasonal march of both the southern westerlies and sea ice, I wonder if it wouldn't be better to either concentrate on a season, or investigates the seasons of minimum/maximum wind/sea ice. Further, when comparing simulated sea ice to reconstructions, this distinction could also be made, as done in Roche et al 2012 using the same sea ice reconstructions and the PMIP2 data. This aspect does need to be at least clarified before publication.

5. The introduction is clearly written, especially on the westerlies, but missed the above reference on previous work on LGM sea ice model-data comparison.

Minor:

- Section 3.1 first paragraph, discussion of Table 2. This discussion mentions the mean and the median of jet changes, but the table itself doesn't give the median. It might actually be easier for the reader to present these results with graphics rather than a table.

- Section 3.3, line 182: are the figures of 7 and 10° in latitude for the Atlantic and Pacific sectors?

- Section 4, line 307: "increase in U". "U" could be replaced by its full meaning for this conclusion section.

- Figure 1, 2nd line, l.h.s. panel: what is the vertical line at latitude ∼-56°S? The information on which time averaging (annual mean?) is also missing (and is missing in the captions of Fig 4, 5, 6).

———————————————————

---

## Author Comment (AC1) · 29 Jul 2016

*General Comments*

This manuscript addresses the issue of how the southern hemisphere westerly winds changed during the last glacial maximum. Changes in the winds have been invoked as a significant player in the glacial-interglacial $CO_2$ variations (the so called "wind-hypothesis"), hence there has been in recent years a number of papers trying to gain insight on this issue from a modelling perspective. A conclusive support or disapproval of the "wind hypothesis" has remained elusive. This paper uses the same existing simulations reported in at least 3 other papers (Chavaillaz et al, 2013, Rojas, 2013 and Liu et al, 2015), in this sense there is a feeling that the results presented are not very new. The main hypothesis for explaining the variety of wind responses to LGM conditions is the sea-ice response of the model and the state dependency. The sea-ice hypothesis has also been invoked in other papers, but the complete analysis of the relationship with the presented day simulation of the winds is novel. In this respect the paper is a contribution towards settling this longstanding issue. However there are at least two important papers that have not been cited: Liu et al 2015: The de-correlation of westerly winds and westerly wind stress over the Southern Ocean during the Last Glacial Maximum. Journal of Climate At the end we are interested in understanding if changes in winds can be responsible for part of the glacial-interglacial atmospheric $CO_2$ variations, by helping to outgass $CO_2$ from the southern ocean. This aspect is addressed in Liu et al 2015. And in this paper this issue is left at a secondary plane. Also, given that sea-ice is diagnosed as a critical component of the LGM response of the system in the southern hemisphere, authors should discuss: Roche, D. M., Crosta, X., & Renssen, H. (2012). Evaluating Southern Ocean sea-ice for the Last Glacial Maximum and pre-industrial climates: PMIP-2 models and data evidence. Quaternary Science Reviews, 56(C), 99–106. http://doi.org/10.1016/j.quascirev.2012.09.020  This paper is PMIP2, so it would be interesting to include in the analysis presented in this paper PMIP2 simulations as well (Rojas 2013, shows the zonal mean winds in PMIP2 and PMIP3).

We thank the reviewer for pointing us towards these papers. References to the Roche et al 2012 and Liu et al 2015 work are now included in the manuscript.

Although the title refers to the conclusion that sea-ice must have been responsible for shifting the SHW poleward during the LGM, the discussion in the paper is too much centered on the modelling errors and how those errors have therefore hindered to date to use models to prove or disprove the "wind-hypothesis". I would re-emphasis the final objective which is to know if the winds where weaker, stronger or shifted during the LGM compared to present day, or pre-industrial climate. Paper requires revisions.

*Specific Comments*

Line 66: is PI defined?
Line 4, in the abstract, and now added here.

Line 69: include reference to Liu et al 2015

Done.

Line 113: include: "..global Ocean circulation and hence CO2 exchange.."
Done.

Line 135: Why do you only analysis annual means, when literature seems to indicate that summer sea-ice is important?
The 15% concentration contour represents the winter sea ice edge. LGM-PI changes in wintertime, or maximum, sea ice extent change are known with more certainty than summertime changes. The position of the wintertime sea ice edge also appears to be have more influence over SWW than the summer edge position (*e.g.* Kidston and Gerber, 2010).

Lines 153-155: can you put those values in a table?
The values are placed on Figure 2. Here, they appear in context with the same quantities.

Line 178: Does the Geresonde paper indicate summer/winter differences?
There is some information on summer/winter differences, but the summer edge is poorly known, compared with the winter edge.

Line 203: change 0 hPa to top level?
Done.

Line 217-218: "…poleward shift in the jet in both models, especially in MRI-CGCM3".
Done.

Line 261: but can only be captured if the PI jet position is accurately simulated.
Done.

Line 262: include discussion of real world!
This sentence has been amended.

Lines 265-268: discuss implication for CO2. In section 4: Can you discuss more on the implication of your finding on CO2 variations?

We comment in the introduction that "the effects on ocean circulation and biology are complex and non-linear, with competing effects from physical and biological carbon pumps." Calculating implications for CO2 change are not within the scope of this work; it is difficult to make any useful statement on CO2 impact of a specified wind shift.

How reliable is the sea-ice reconstruction

The Gersonde et al reconstruction of wintertime sea ice extent seems to be quite robust, but the summer sea ice extent position is not so well known. References to the Roche et al 2012 paper on sea ice are also included in the introduction and conclusion.

**Anonymous Referee #2**

Sime et al. use nine different CMIP5-PMIP3 preindustrial and LGM pairs to investigate the changes in Southern Hemisphere jet position and intensity and how would these changes related to the preconditioned jet location (state dependency) and sea ice expansion. This study is established on previous work by the author (i.e. Sime et al., 2013) and many other works by Kidston and Gerber (2010), Chavaillaz et al. (2013), Bracegirdle et al. (2013) .etc. and further suggest that sea ice being an important factor for the deglacial changes in Southern Hemisphere jet. It is, however, a bit weak on the discussion on how this study agree/disagree from previous studies and basin scale detail.

*Specific Comments:*

1. Do we confident on the actual condition of LGM SH jet condition regardless of mutil-model mean suggests on no significant changes? If positive, then the conclusion that sea ice expansion holding jet in its present day position would valid otherwise the explanation is only for modeling perspective.

The manuscript aims to improve our mechanistic understanding of what drives PI to LGM SH wind changes. It should be useful for the analysis of future model experiments, and be of interest to meteorologists and observationalists, as well as climate modellers. We are confident of our conclusions, so in answer these results should apply to the real world.

2. In line 117, "data is regridded to a consistent 0.1∘ resolution before these calculation are performed." I understand this is to separate the jet latitude between runs but wonder if it is legitimate to so. This is a one to twenty scaling after all as most of the model simulations here has a spatial resolution of 2.5∘. One generally would not interpolate a T42 simulation to T106. By reading Table 2, it should do the job by interpolating data down to 0.5∘. Or it would be nice to show the conclusion is not resolution (aka interpolation) dependent.

We recalculated the results using a 0.5° resolution for the regridding, and find that it make no significant difference. Since regridding to 0.1 seems to be a more standard practice in the literature, and the regridding interval seems not to influence results, we leave the interval at 0.1°.

3. Section 3.1 describes the state dependency in PI-LGM changes. Can the author comment on why these results being quite different from Bracegirdle et al., (2013)? Bracegirdle et al. (2013) suggest strong dependency of jet over Pacific basin in warming scenario from PI to future condition while this study suggest much weaker state dependency in Pacific. Would this related to the different simulated sea ice and temperature conditions between PI-LGM and RCP-PI? This might further support the argument in Section 3.2.1 and 3.2.2.

The main explanation that we suggest is that this is a consequence of the larger sea ice extents in the LGM and pre-industrial simulations compared to present day climate and future warming scenarios. As mentioned in Section 3.2.1, changes in ice extent that occur at

lower latitudes are generally in closer proximity to the mid-latitude jet and therefore influence it more strongly. The magnitude of sea ice change therefore becomes a key influence on jet position change and has greater potential to disrupt the state dependency that is seen in future projections from present-day conditions (e.g. Kidston and Gerber, 2010; Bracegirdle et al., 2013).

4. Continue from previous comment, section 3.2 discuss the impact of sea ice. Is it possible to calculate the percentage of variation explained by state dependency and sea ice separately? In other word, which factor represents a stronger control over PI-LGM jet variability?

We do not think this is possible. In coupled models it is very challenging to estimate the extent to which the sea ice is causing the wind change and vice versa.

5. In reading Figure 5 and 6, it shows a non-proportional changes between temperature gradient structure and U wind changes. Can the author comment on this? For example, COSMO and MPI-ESM both show substantial changes in temperature gradient while MPI-ESM simulate none changes in zonal wind.

Reviewer #2 identifies that the details of the link between wind and temperature gradient change are not proportional. Although, broadly, larger increases in temperature gradient over the troposphere give larger westerly wind increases over the troposphere, this is actually because increases in horizontal temperature gradient lead to increases in westerly wind with height. These wind changes are generally quite small near the surface and either increase or decrease with height depending on the sign of the temperature gradient change. We have now alluded to this point in the main text, whilst taking care not to obscure the first-order picture.

6. Suggestion: The authors mention in the manuscript that the changes in sea ice might be important in determining LGM SH jet changes. It can be verify and support by comparing simulations with different sea ice extent, say LGM-PI-RCPs from extensive sea ice to sea ice free.

One of the main findings of this manuscript is that sea ice changes have more impact when they are positioned close to the location of the wind jet. Because warm period simulations, such as the RCP simulations, have a sea ice edge which is far from the jet, changes in sea ice are unlikely to influence the SH winds through the same mechanisms. We add references to Bracegirdle et al 2013 and clarify this point in the text.

7. Very minor: in line 171 and line 300, as far "north" as 57◦S, is this a typo of "south" relative to 47◦S
Thank you - corrected.

**Anonymous Referee #3**

*General comments:*

This manuscript investigates the Last Glacial Maximum (LGM) southern westerlies, a subject of high interest due to their possible control of the atmospheric CO2, whose LGM level is still not fully understood, but also due to their importance in atmosphere-ocean exchanges and ocean circulation. The southern westerlies are predicted to shift poleward under climate warming, and would therefore be expected to shift equatorward during the cooler-than-present LGM. Previous work has shown that this is not the case. This manuscript further analyses the possible implication of sea ice changes in LGM southern jet changes. It is based on the PMIP3-CMIP5 data, which is the most recent data available from several state-of-the-art models run with the same boundary conditions. For sea-ice, this data is compared to the reconstructed data set of Gersonde et al (2005). The authors show that models which are sufficiently accurate for the present climate in their representation of the jet stream, i.e. for which the position of the jet stream maximum is not too much biased northward, and simulate a realistic sea-ice extension from present to LGM, can show a poleward shift of the jet stream for the LGM compared to present. Although the southern westerlies had been investigated in previous publications, this work goes one step further by 1/ clearly showing the impact on present day biases on the LGM – present changes and 2/ showing the role of sea-ice for those models which simulate realistic westerlies for the present situation. The manuscript is clearly written and illustrated, I am therefore in favour for its publication in Climate of the Past, after the corrections and improvements listed below.

*Suggested improvements:*

1. Since there are not so many models which have provided the complete data set for this analysis (in particular sea ice concentration) the authors may want to encourage the corresponding modelling groups to submit their data to the CMIP5-PMIP3 data base. This might increase the numbers used to infer the relationship between the jet shift and sea-ice edge or SSTs on Fig. 7.

It would indeed be very useful if there were more CMIP6 modelling groups who submitted paleoclimate modelling simulations for the next edition of PMIP. In CMIP6 we very much hope that this will be the case. However for CMIP5, it is unfortunately now too late in the cycle for groups to submit additional simulations or variables.

2. Another way to increase the number of models, by using PMIP2 data, could also help, but these models being of an older generation might show even stronger biases?

We think the reviewer is correct to imply that it is preferable to use the most up-to-date CMIP5 models.

3. Finally, would the study of the yearly fields help and quantify the uncertainty of the results shown on Fig. 7 (regression between jet position and sea ice edge)? e.g. by using bootstrapping to add information on the variability of the results?

It is not possible to quantify the uncertainty use the year-to-year variability of sea ice and jet position to increase the number of points in Figure 7. Ferreira et al (2015) work showed that the ocean-ice-atmosphere system responds to wind changes in different ways on annual

compared to multi-annual timescales, at least for most models. For example, see their figure 12 - in Ferreira, D., J. Marshall, C. M. Bitz, S. Solomon, A. Plumb, 2015: Antarctic Ocean and sea ice response to ozone depletion: a two timescale problem. J. Climate, 28, 1206-1226.

4. One aspect of the manuscript I am very confused with is the seasonal/annual averaging of the data. On the one hand, section 2.1 specifies that "jet diagnostics are calculated for the climatological maximum in the Southern Hemisphere 850 hPa wind component" but the season (or month?) is not specified. On the other hand, Figures 2, 3 and 4 show annual means of winds and sea ice. Given the large seasonal march of both the southern westerlies and sea ice, I wonder if it wouldn't be better to either concentrate on a season, or investigates the seasons of minimum/maximum wind/sea ice. Further, when comparing simulated sea ice to reconstructions, this distinction could also be made, as done in Roche et al 2012 using the same sea ice reconstructions and the PMIP2 data. This aspect does need to be at least clarified before publication.

This is an issue with the terminology, we used climatological maximum, when we should have used annual mean. This is now fixed throughout the manuscript.

5. The introduction is clearly written, especially on the westerlies, but missed the above reference on previous work on LGM sea ice model-data comparison.

The Roche et al 2012 reference has now been added to the introduction.

*Minor:*

- Section 3.1 first paragraph, discussion of Table 2. This discussion mentions the mean and the median of jet changes, but the table itself doesn't give the median. It might actually be easier for the reader to present these results with graphics rather than a table.

It is possible to calculate the median value from the table results, though having the figure in the text may be is helpful, it would be more difficult to calculate these results from a figure.

- Section 3.3, line 182: are the figures of 7 and 10deg in latitude for the Atlantic and Pacific sectors?

This is now clarified in the text.

- Section 4, line 307: "increase in U". "U" could be replaced by its full meaning for this conclusion section.

Now replaced.

- Figure 1, 2nd line, l.h.s. panel: what is the vertical line at latitude ~-56S? The information on which time averaging (annual mean?) is also missing (and is missing in the captions of Fig 4, 5, 6).

Caption problems fixed.

---

## Referee Report (RR1)

Review 2 of Sime et al, 2016

I have read both the comments of the three reviewers of this paper as well as the answer to these comments by the authors.
All points raised by the reviewers seem to have been taken into account, hence the paper can be accepted. There are still some very minor points I would like to see addressed:

1. I find it odd to call the "Southern westerly winds" "Southern Ocean westerly wind jet"….mixing up oceanic and atmospheric features. The winds are in the atmosphere, blowing over "southern ocean" latitudes. I would change this both in the abstract (line 1) and Introduction (line 26) and elsewhere.
2. Discussion of lines 24-28: I would include reference to work that do indicate a correlation or relationship between the deglacial CO2 evolution and Antarctic temperatures, that let to the hypothesis that there must be Southern Ocean control.
3. Summary and conclusions: lines 294-297: say something about the Pacific.